

# Panel strain of *Klebsiella pneumoniae* for beta-lactam antibiotic evaluation: their phenotypic and genotypic characterization

Roshan Dsouza[1], Naina Adren Pinto[1,2], InSik Hwang[1,2], YoungLag Cho[3], Dongeun Yong[1], Jongrak Choi[1], Kyungwon Lee[1] and Yunsop Chong[1]

[1] Department of Laboratory Medicine and Research Institute of Bacterial Resistance, Yonsei University College of Medicine, Seoul, South Korea
[2] Brain Korea 21 PLUS Project for Medical Science, Yonsei University, Seoul, South Korea
[3] LegoChemBioScience Inc, Daejeon, South Korea

## ABSTRACT

*Klebsiella pneumoniae* is responsible for numerous infections caused in hospitals, leading to mortality and morbidity. It has been evolving as a multi-drug resistant pathogen, acquiring multiple resistances such as such as horizontal gene transfer, transposon-mediated insertions or change in outer membrane permeability. Therefore, constant efforts are being carried out to control the infections using various antibiotic therapies. Considering the severity of the acquired resistance, we developed a panel of strains of *K. pneumoniae* expressing different resistance profiles such as high-level penicillinase and AmpC production, extended spectrum beta-lactamases and carbapenemases. Bacterial strains expressing different resistance phenotypes were collected and examined for resistance genes, mutations and porin alterations contributing to the detected phenotypes. Using the Massive parallel sequencing (MPS) technology we have constructed and genotypically characterized the panel strains to elucidate the multidrug resistance. These panel strains can be used in the clinical laboratory as standard reference strains. In addition, these strains could be significant in the field of pharmaceuticals for the antibiotic drug testing to verify its efficiency on pathogens expressing various resistances.

## INTRODUCTION

Over the last three decades, we have observed increased occurrence of multidrug-resistant *Enterobacteriaceae*. These are constantly evolving as resistant pathogens posing the serious problems in the choice of an appropriate antibiotic treatment in the hospital settings (*Davies & Davies, 2010*). *Klebsiella pneumoniae* are emerging as one of the primary opportunistic pathogens causing a significant amount of mortality and morbidity (*Peleg & Hooper, 2010*) in hospitals, causing urinary tract infections, pneumonia, bloodstream infections, surgical site infections, and meningitis (*Davis et al., 2015*; *Ko et al., 2002*; *Pereira et al., 2015*; *Ahn et al., 2016*; *Oh et al., 2015*). Over the years, it has evolved to be multi-drug resistant, showing high resistance to extended spectrum beta-lactam (ESBL), fluoroquinolones, aminoglycosides and even the last resort 'carbapenems' (*Fair & Tor, 2014*).

Corresponding author
Dongeun Yong, DEYONG@yuhs.ac

To validate these resistances various mechanisms have been illustrated, such as high level production of AmpC $\beta$-lactamase and penicillinase, acquisition of genes encoding for Extended Spectrum Beta-Lactamase (ESBL) or carbapenemase, change in the membrane permeability or high level expression of efflux pump systems (*Blair et al., 2015*; *Tsai et al., 2013*). *K. pneumoniae* producing AmpC $\beta$-lactamase, which is a plasmid mediated, has become major therapeutic challenge due to their resistance to cephalothin, cefazolin, cefoxitin and $\beta$-lactam inhibitor combinations (*Gonzalez Leiza et al., 1994*; *Horii et al., 1993*; *Jenks et al., 1995*). Induction and over expression of these enzymes has been linked with peptidoglycan recycling involving AmpD-AmpR-AmpC gene regulatory networks in *enterobacteriaceae* (*Anitha et al., 2015*; *Guerin et al., 2015*). ESBL are also plasmid-mediated, which are complex and rapidly evolving enzymes that hydrolyze third- generation cephalosporins and aztreonam but are inhibited by clavulanic acid (*Rawat & Nair, 2010*). There are more than 200 ESBLs have been discovered originating from more than 30 different countries (http://www.lahey.org/studies/). Previous studies indicate that *K. pneumoniae* consists of several large plasmids which carry vast number of ESBLs and carbapenemases genes along with the genes for resistance to aminoglycosides, trimethoprim, sulphonamides, tetracyclyclins and chloramphenicol (*Conlan et al., 2016*; *Paterson, 2000*; *Tokajian et al., 2015*). These plasmids are encoded with wide variety of adhesin-related gene clusters, mobile genetic elements such as conjugation transfer genes such as *tra*, transposons and insertion sequences which help in adaptive evolution, horizontal gene transfer, thus spreading the antibiotic resistance from one species or genera to another (*Liu et al., 2012*; *Rafiq, Sam & Vaidyanathan, 2016*; *Ramos et al., 2014*). Increased resistance to carbapenems and glycylcycline has also been facilitated by alterations in membrane permeability/potential or altered efflux pumps (*Cannatelli et al., 2014*; *Filgona, Banerjee & Anupurba, 2015*; *He et al., 2015*; *Seecoomar, Marmol & Kwon, 2013*). Efflux pumps such as AcrAB, KexD, KdeA, KmrA, kpnEF and kpnGH confer resistance to wide spectrum of antimicrobial agents in *K. pneumoniae* (*Ogawa et al., 2006*; *Ogawa et al., 2012*; *Padilla et al., 2010*; *Ping et al., 2007*; *Srinivasan & Rajamohan, 2013*; *Srinivasan et al., 2014*).

Rapid development in the field of massive parallel sequencing (MPS) has enabled the clinical microbiology laboratory to gain better insights into understanding the bacterial resistance (*Goldberg et al., 2015*; *Koser et al., 2012*; *Koser, Ellington & Peacock, 2014*). Considering the severity of incidence of *K. pneumoniae* infections we have constructed a panel of strains, consisting of different resistance phenotypes. We have also elucidated the relationship between displayed phenotypes with its corresponding genotypic profile using MPS technology. This study was primarily designed for the preliminary screening of the efficacy of antimicrobial compounds on *K. pneumoniae* expressing different resistance phenotypes in pharmaceutical industry. These strains can be distributed to hospitals and institutions undertaking research on antimicrobial resistance.

## MATERIALS AND METHODS

### Bacterial isolates

*K. pneumoniae* isolates were collected during 2009 to 2013 from a tertiary care university affiliated hospital in Seoul, Korea. Bacteria identification was performed using VITEK 32 GN system (BioMérieux, Marcy l'Etoile, France), and was confirmed using the direct colony method with MALDI-TOF MS (Bruker Daltonics, Bremen, Germany). These isolates were screened for specific phenotypes such as high level acquired penicillinase, over produced AmpC, ESBL, and carbapenemase based on Bonnet R, et al. in "Antibiogram" (*LeClercq, Courvalin & Rice, 2010*) and "From antibiogram to prescription" book (*François et al., 2004*) from the hospital database.

### Susceptibility tests and MIC determinations

Around 3,000 *K. pneumoniae* isolates were shortlisted for antibiotic susceptibility test. This was performed by the disc diffusion method using piperacillin, ampicillin, piperacillin/tazobactam, ceftazidime, cefepime, imipenem, meropenem, ciprofloxacin, ceftazidime-clavulanate, ampicillin-sulbactam, aztreonam on Mueller-Hinton agar. Further confirmatory tests were performed for the above antibiotics by agar-dilution technique. All the results were interpreted according to the Clinical and laboratory standards Institute (CLSI) guidelines (2015).

ESBL isolates were sorted out using double disk synergy test using cefepime and clavulanate and Hodge test using cefoxitin. Imipenem and EDTA double disk synergy along with Hodge test were used to select the carbapenemase-producing isolates. High level AmpC were selected using ertapenem and aminophenyl boronic acid (APBA) double disk synergy test.

### Resistance gene confirmation

Phenotypically confirmed isolates were cultured overnight and suspended in distilled water, heated at 95 °C for 10 min. The suspension was centrifuged for 1 min at 6,000 rpm and the supernatant was used as a DNA template. Primers were designed for resistance genes (Table S1) and ordered from Macrogen (Seoul, Korea) and PCR was performed using accupower PCR premix (Bioneer, Korea).

### DNA isolation

Random strains were picked from each resistance phenotypes and cultured overnight. Both genomic and plasmid DNA were isolated using Wizard genomic DNA purification kit (Promega, Madison, WI, USA) with little modification to the manufacturer's protocol, and Qiaprep spin miniprep kit (Qiagen, Hilden, Germany), respectively. DNA concentration was estimated through Qubit dsDNA BR assay kit (Molecular Probes, Eugene, OR, USA).

### Ion Torrent PGM sequencing

Whole genome library was performed using Ionplus fragment library kit (Life Technologies, Carlsbad, CA, USA). Emulsion PCR was carried out using the IonOnetouch 200 Template kit v2 DL (Life Technologies, Carlsbad, CA, USA) according to the manufacturer's

instructions. Sequencing of the libraries was carried out on a 318 chip using the Ion Torrent PGM system and Ion Sequencing 200kit (Life Technologies, Carlsbad, CA, USA).

## Sequence assembly, annotation, multilocus sequence typing (MLST) and resistome analysis

Reads from Ion Torrent PGM system were assembled using MIRA plug-in available in Torrent suite software. Annotations were performed using the RAST annotation pipeline with manual scrutiny. Genomic analysis was performed using Geneious pro 8.0 (http://www.geneious.com; *Kearse et al., 2012*). Resistance genes were screened using Resfinder (*Zankari et al., 2012*; https://cge.cbs.dtu.dk/services/ResFinder/) and they were further verified using NCBI BLAST. All the references used to annotate the resistance genes are listed in Table S2. Bacterial typing was performed using online tool MLST 1.8 (*Zankari et al., 2013*; https://cge.cbs.dtu.dk/services/MLST/).

## OMP detection

Bacterial cells were grown in high-osmolarity MHB to the logarithmic phase and lysed by sonication at 18–20% amplitude for 2 × 30s cycles, each comprised 6 × 5s sonication steps separated by 1s of no sonication, and 30s of no sonication between the two cycles. Unbroken cells were separated using centrifugation at 3,000 g for 5 min and Outer membrane proteins (OMP) were extracted with Sodium lauroyl sarcosinate and recovered by ultracentrifugation, as described previously (*Hernandez-Alles et al., 1999*). The OMP profiles were determined using SDS-PAGE using Mini-Protean TGX gels followed by coomassie blue staining (Bio-Rad, Hercules, CA, USA). Additionally, OMP's were detected using Matrix-Assisted Laser Desorption-Time of Flight Mass Spectrometry on Tinkerbell LT (ASTA, Suwon, Korea) as described in *Cai et al. (2012)*. All the experiments were repeated thrice to check the reproducibility of the results.

## RESULTS AND DISCUSSIONS

Among 3,000 *K. pneumoniae* collection, we could finally select eighteen isolates showing typical phenotypes i.e. six ESBL producing, nine carbapenemase producing, one isolate expressing High level penicillinase, one high level AmpC $\beta$-lactamase producing and one wild-type strain susceptible to antibiotics except ampicillin. MIC's of these strains are illustrated in Table 1 and Table S3. YMC2011/8/B10311 (High level acquired penicillinase); YMC2011/7/B774, YMC2013/7/B3993, YMC2011/11/B7578, YMC2011/7/B7207 (ESBL); YMC2010/8/B2027 (High level AmpC $\beta$-lactamase) and YMC2012/8/C631 (Carbapenemase) were sequenced to obtain the complete genotypic and phenotypic correlation (Table 2). The assembly statistics and the annotation overview are indicated in Table 3. Consistent with the previous sequencing studies of *K. pneumoniae*, the genomic size was about 5–9 -mbp sequences with an average G + C content of 57%. More than 650,000 high quality reads were assembled to produce the draft genomes of an average 30 fold coverage (Table S4). There are more than 5,000 predicted protein coding sequences and 96 RNA's within the genomes of sequences panel strains. Table S5 indicates the number of subsystems which reveal the number of genes involved in specific biological process. To

**Table 1** Selected list of panel strains and its MIC.

| Strains | | PIP | CAZ | FEP | IMI | MER | CAZ/CLV | FOX | AMP | SAM |
|---|---|---|---|---|---|---|---|---|---|---|
| **ESBL** | MLST | R | R | R | S | S | S | S | R | V |
| YMC2011/7/B774 | 551 | 256(R) | 256(R) | 128(R) | 0.25(S) | 0.25(S) | 1(S) | 32(R) | 256(R) | 64(R) |
| YMC2013/7/B3993 | 11 | 256(R) | 256(R) | 64(R) | 0.25(S) | 0.25(S) | 1(S) | 32(R) | 256(R) | 128(R) |
| YMC2011/7/B7207 | 711 | 256(R) | 64(R) | 16(I) | 0.25(S) | 0.25(S) | 1(S) | 32(R) | 256(R) | 64(R) |
| YMC2011/11/B7578 | 11 | 256(R) | 256(R) | 64(R) | 64(R) | 1(S) | 1(S) | 128(R) | 256(R) | 64(R) |
| **High level Ampc β-lactamase** | | R | R | S | S | S | R | R | R | R |
| YMC2010/8/B2027 | 517 | 256(R) | 256(R) | 2(S) | 0.5(S) | 0.25(S) | 64(R) | 256(R) | 256(R) | 128(R) |
| **Carbapenemase** | | R | R | R | R | R | R | R | R | R |
| YMC2012/8/C631 | 354 | 256(R) | 256(R) | 32(R) | 64(R) | 64(R) | 32(R) | 256(R) | 256(R) | 128(R) |
| **High level acquired penicillinase** | | R | S | S | S | S | S | S | R | R |
| YMC2011/8/B10311 | 17 | 256(R) | 2(S) | 4(S) | 0.5(S) | 2(I) | 2(S) | 8(S) | 256(R) | 128(R) |

Notes.

MLST, Multilocus sequence typing; R, Resistant; I, Intermediate; S, susceptible; PIP, piperacillin; PIP/TZ, piperacillin-tazobactam; CAZ, ceftazidime; FEP, cefepime; AZT, aztreonam; IMI, imipenem; MER, meropenem; CIP, ciprofloxacin; CAZ/CLV, ceftazidime-clavulanate; FOX, cefoxitin; AMP, ampicillin; SAM, Ampicillin/Sulbactam.

characterize further, SDS-PAGE for detection of OMP analysis was performed for these strains (Fig. S1), which was confirmed using MALDI-TOF Figures S2 and S3 indicates the alignment of OmpK35 and OmpK36 genes of panel strains including their promoter regions. The draft genome sequences of strains YMC2011/8/B10311, YMC2011/7/B774, YMC2013/7/B3993, YMC2011/7/B7207, YMC2011/11/B7578, YMC2010/8/B2027 and YMC2012/8/C631 have been deposited at DDBJ/ENA/GenBank under the accession LYPQ00000000, LYPS00000000, LDWV00000000, LYPU00000000, LYPT00000000, LYPV00000000 and LYPW00000000, respectively.

## High level acquired Penicillinase

*K. pneumoniae* YMC2011/8/B10311 was susceptible to piperacillin-tazobactam, ceftazidime, cefepime, imipenem, meropenem, ciprofloxacin, ceftazidime-clavulanate and cefoxitin but resistant to piperacillin, ampicillin, and ampicillin-sulbactam. WGS analysis shows the presence of $bla_{SHV-11}$ and $bla_{TEM-1}$ genes. Resistance to the piperacillin, ampicillin and ampicillin-sulbactam are due to hyperproduction of penicillinase TEM-1and SHV-11 beta- lactamase. OmpK35 gene was present while OmpK36 gene expression was truncated or terminated due to the mutations present as observed in the WGS. It was also confirmed using SDS-PAGE, which revealed OmpK35 porin alone.TEM-1 beta-lactamase offers resistance to $\alpha$- amino and - carboxy - penicillins in *E. coli* and *Enterobacteriaceae*. Generally, in high level acquired penicillinase strains, there was increased production of TEM-1, which can be inhibited by piperacillin-tazobactam efficiently than the ampicillin-sulbactam combination (*Livermore & Seetulsingh, 1991*), as confirmed above.

Dsouza et al. (2017), *PeerJ*, DOI 10.7717/peerj.2896

**Table 2 Resistome analysis of the selected panel strain.**

| | β-Lactam | | | | | | | | | | | Aminoglycoside | | | | | | | Quinolone | | | | | Phenicol | | TET | TMP | Sulfonamide | | Porin | |
|---|---|---|---|---|---|---|---|---|---|---|---|---|---|---|---|---|---|---|---|---|---|---|---|---|---|---|---|---|---|---|---|
| | DHA-1 | CMY-2 | CTX-M-15 | IMP-1 | OXA-1 | OXA-9 | SHV-11 | SHV-12 | SHV-158 | SHV-187 | TEM-1 | aac(6')-Ib | Aac(6')-IIa | aadA1 | aadA2 | armA | strA | strB | QnrB66 | QnrB4 | aac(6')Ib-cr | oqxA | oqxB | catA2 | catB3 | tet(A) | dfrA14 | sul1 | sul2 | OmpK35 | OmpK36 |
| **ESBL** | | | | | | | | | | | | | | | | | | | | | | | | | | | | | | | |
| YMC2011/7/B774 | | | ● | | ● | | ● | | | | ● | ● | | | | | ● | ● | ● | | ● | | ● | | ● | ● | ● | | ● | ● | |
| YMC2013/7/B3993 | | | ● | | | ● | | ● | | | ● | ● | ● | ● | | | ● | ● | ● | | | ● | ● | | | | ● | | ● | | ● |
| YMC2011/7/B7207 | | | ● | | ● | | | | | ● | ● | ● | | | | | ● | ● | ● | | ● | ● | ● | | ● | ● | ● | | ● | ● | |
| YMC2011/11/B7578 | ● | | | | | | | ● | ● | | | | | | ● | ● | ● | ● | ● | | | ● | ● | | | | | | ● | ● | |
| **High level Ampc β-lactamase** | | | | | | | | | | | | | | | | | | | | | | | | | | | | | | | |
| YMC2010/8/B2027 | ● | ● | | | | | ● | | | | ● | ● | | | | | ● | ● | ● | ● | | | | ● | | | ● | ● | | ● | ● |
| **Carbapenemase** | | | | | | | | | | | | | | | | | | | | | | | | | | | | | | | |
| YMC2012/8/C631 | ● | | | ● | | | | | | | ● | ● | | | | | ● | ● | | | | | | | | | | ● | ● | ● | |
| **High level acquired penicillinase** | | | | | | | | | | | | | | | | | | | | | | | | | | | | | | | |
| YMC2011/8/B10311 | | | | | | | ● | | | | ● | | | | | | | | | | | ● | ● | | | ● | ● | | | ● | |

**Table 3** Assembly statistics and annotation overview of the panel strains.

| Strains | Size(bp) | Assembled reads(bp) | Coverage | Contigs | N50(bp) | N90(bp) | N95(bp) | GC(%) | Sub systems | Coding sequences | RNA |
|---|---|---|---|---|---|---|---|---|---|---|---|
| **ESBL** | | | | | | | | | | | |
| YMC2011/7/B774 | 5,459,074 | 678,178 | 26.27 X | 90 | 121,292 | 35,800 | 22,360 | 57.3 | 587 | 5175 | 107 |
| YMC2013/7/B3993 | 5,908,460 | 851,171 | 30.63 X | 188 | 115,815 | 21,473 | 7,508 | 57 | 590 | 5714 | 114 |
| YMC2011/7/B7207 | 5,307,765 | 839,982 | 31.96 X | 61 | 209,753 | 55,148 | 33,204 | 57.5 | 581 | 5039 | 113 |
| YMC2011/11/B7578 | 5,635,222 | 693,699 | 30.89 X | 104 | 143,377 | 32,570 | 21,435 | 57.2 | 588 | 5488 | 112 |
| **High level Ampc $\beta$-lactamase** | | | | | | | | | | | |
| YMC2010/8/B2027 | 5,848,366 | 1,054,071 | 38.08 X | 132 | 123,629 | 29,379 | 18,824 | 56.5 | 592 | 5911 | 111 |
| **Carbapenemase** | | | | | | | | | | | |
| YMC2012/8/C631 | 5,879,989 | 909,372 | 32.32 X | 233 | 84,557 | 10,295 | 5,587 | 56.7 | 589 | 5727 | 112 |
| **High level acquired penicillinase** | | | | | | | | | | | |
| YMC2011/8/B10311 | 5,478,035 | 739,898 | 26.42 X | 78 | 145,826 | 36,572 | 26,851 | 57.3 | 588 | 5242 | 96 |

### Extended spectrum beta-lactamase

ESBL producing strains were phenotypically confirmed using the double disk and Hodge test. Most of these strains belonged to ST11, which are predominant in Korea since 2010 (*Ko et al., 2010*).

#### YMC2011/7/B774

This strain was susceptible to imipenem, meropenem and ceftazidime-clavulanate, intermediate to piperacillin-tazobactam but resistant to piperacillin, cefepime, cefoxitin, ciprofloxacin, ampicillin, and ampicillin-sulbactam. The resistance is due to the presence of $bla_{CTX-M15}$ along with $bla_{OXA-1}$. The $bla_{OXA-1}$ gene has been frequently found to be associated with genes encoding ESBL's. This, along with the OmpK36 porin loss can cause reduced susceptibility to cefepime (*Beceiro et al., 2011*; *Torres et al., 2016*). $bla_{OXA1}$ was found in the following genetic environment, $IS26$-$catB4$-$bla_{OXA-1}$-$aac(6')$-$Ib$-$cr$-$IS26$. The presence of multiple aminoglycoside resistance genes such as $aac(6')Ib$-$cr$, $strA$, $strB$, $QnrB66$ and $oqxB$ have offered resistance to ciprofloxacin. Cefoxitin resistance was mediated by loss of porins which is well described in *K. pneumoniae* strains (*Ananthan & Subha, 2005*).

#### YMC2013/6/B3993

This multidrug-resistant strain was unique because it contained multiple copies of ESBL gene ($bla_{SHV-12}$). This was susceptible to imipenem, meropenem and ceftazidime-clavulanate but resistant to piperacillin, piperacillin-tazobactam, ceftazidime, cefepime, cefoxitin, ciprofloxacin, ampicillin, and ampicillin-sulbactam. In-depth analysis of the strain revealed 2 copies of $bla_{SHV-12}$ and one copy of $bla_{CTX-M-15}$ genes. In-addition, we also found one copy of $bla_{OXA-9}$ and three copies of $bla_{TEM-1}$. The strain belonged to ST11.The insertion of *Tn1331*was detected, consisting of $bla_{OXA-9}$, $bla_{TEM-1}$, $aac(6'$-$)Ib$-$cr$ and *aadA1* genes. Ciprofloxacin resistance was notably high (MIC 128 mg/dl), which was due the additive effect of both quinolone resistance-determining regions

(QRDR) and plasmid-mediated quinolone resistance (PMQR). Mutations in QRDR were noticed at Ser83Ile and Ser80Ile in *gyrA* and *parC* genes, respectively. PMQR analysis indicated the presence of *aac(6′)-Ib-cr* and *qnrB*, with efflux pumps *oqxA* and *oqxB*. In silico analysis of the strain confirmed the presence of OmpK35 and OmpK36 porins. However, the OmpK35 gene has been interrupted by *IS1*, thus providing resistance to cefoxitin (*Ananthan & Subha, 2005*; *Palasubramaniam et al., 2007*). OmpK36 belong to the OmpK36_v1 variant with the amino acid substitution at Arg357His with a nucleotide substitution from A to T at −10 box. Reduced susceptibility to cefepime was due to the multiple copies of $bla_{TEM-1}$ along with truncated OmpK36 porin, which is consistent with the previous studies (*Beceiro et al., 2011*).

### YMC2011/7/B7207

This strain was susceptible to imipenem, meropenem and ceftazidime-clavulanate, intermediate to piperacillin-tazobactam and cefepime but resistant to piperacillin, ceftazidime, cefoxitin, ciprofloxacin, ampicillin, and ampicillin-sulbactam. Similar to the strain YMC2011/7/B774, this strain has $bla_{CTX-M15}$ and $bla_{OXA-1}$ along with the $bla_{SHV-187}$. Reduced susceptibility to cefepime is due to the $bla_{OXA-1}$ (*Torres et al., 2016*). The presence of multiple aminoglycoside resistance genes such as *aac(6′)Ib-cr*, *strA*, *strB*, *oqxA*, *oqxB* and *QnrB66* offered resistance to ciprofloxacin. SDS-PAGE for OMP's revealed the presence of the OmpK35 alone. *OmpK36* gene included several mutations (Table S6) leading to the termination of its expression, presumably leading to cefoxitin resistance.

### YMC2011/11/B7578

This was susceptible to imipenem, meropenem, ceftazidime-clavulanate and intermediate to piperacillin-tazobactam but resistant to piperacillin, ceftazidime, imipenem, cefoxitin, cefepime, ciprofloxacin, ampicillin, ampicillin-sulbactam. The presence of $bla_{SHV-158}$, $bla_{SHV-12}$ along with AmpC gene $bla_{DHA-1}$ explains the resistance to cefoxitin similar to the above strain. Ciprofloxacin resistance was due to the presence of fluoroquinolone resistant genes such as *QnrB4, OqxA and OqxB*. OmpK35 was present and there was a deletion (313G) in OmpK36, which might have caused disruption in its expression leading to cefepime resistance. We could not explain the ceftazidime-clavulanate susceptibility of the strain in spite of the presence of AmpC gene, $bla_{DHA-1}$. We are performing additional experiments to understand this specific phenotype.

## High level AmpC beta-lactamase

*K. pneumoniae* YMC 2010/8/B2027 was found to be susceptible to cefepime, imipenem and meropenem and intermediate to ciprofloxacin but resistant to piperacillin, piperacillin-tazobactam, ceftazidime, ceftazidime-clavulanate, cefoxitin, ampicillin and ampicillin-sulbactam. The resistance phenotype can be because of the presence of AmpC genes i.e. $bla_{DHA-1}$ and $bla_{CMY-2}$. It also carries the broad spectrum beta-lactamase $bla_{SHV-11}$ and penicillinase gene $bla_{TEM-1}$. Ciprofloxacin resistance has been conferred by the presence of *aac(6′)Ib-cr, strA and strB*.

## Carbapenemases

*K. pneumoniae* YMC 2012/8/C631 was found to be susceptible to piperacillin-tazobactam and ciprofloxacin but resistant to piperacillin, ceftazidime, imipenem, meropenem, ceftazidime-clavulanate, cefoxitin, ampicillin and ampicillin-sulbactam. The cefoxitin resistance was due to the presence of $bla_{DHA-1}$. Even though IMP-1 enzyme is known to hydrolyze piperacillin and tazobactam, this strain appeared susceptible. Similar cases have been reported earlier (*Chen et al., 2009*; *Koh, Wang & Sng, 2004*; *Mushtaq, Ge & Livermore, 2004*; *Picao et al., 2013*; *Santella et al., 2010*; *Scheffer et al., 2010*), which may be due to inherent susceptibility to the particular antibiotics. Other resistances are conferred due to the presence of 3 copies of $bla_{TEM-1}$ and $bla_{SHV-11}$. While SDS-PAGE revealed the presence of OmpK35, in silico analysis of *OmpK35* gene showed the insertion of *IS102*, thus affecting antibiotic passage through the membrane.

In addition, as opposed to the findings by *Zankari et al. (2012)* we found few discrepancy in the identification of antimicrobial resistance genes by using Resfinder using the whole genome sequencing, such as few genes were identified as $bla_{LEN-11}$ instead of $bla_{SHV-11}$/$bla_{SHV-12}$ in the above panel strains. Hence the above panel strains needed to be further evaluated for accurate identification. This software consists of database of resistance genes, which helps us to easily identify the resistance mechanism. However, manual scrutiny of the Resfinder results is essential to identify the true antibiotic resistance genes present in the pathogen. Figure S4 indicates the encoded amino acid alignment of the SHV type genes found in the panel strains. $bla_{SHV-11}$ is a broad spectrum $\beta$-lactamase gene, encodes 286 amino acid, in which the mutations $Gly_{234}$-$Serine_{234}$ and $Glu_{235}$-$Lys_{235}$ results in SHV-12 $\beta$-lactamase, $Gln_{31}$-$Leu_{31}$ results in SHV-187 $\beta$-lactamase and $Thr_{54}$-$Ala_{54}$ leads to SHV-158 $\beta$-lactamase.

The advent of NGS in clinical laboratory field has helped us to gain better insights of resistant mechanisms in detail compared to the traditional phenotypic detection. The analysis of WGS will also help us to understand the collective molecular network of pathogen offering the specific MIC with the relevant antibiotics (*Tsai et al., 2013*). In addition, it also plays an important role in clarifying the discrepancy observed due to false negative results generated from existing diagnostic assays (*Koser, Ellington & Peacock, 2014*). These assays mostly target the single resistance mechanism or phenotype, which is not sufficient to understand the complete underlying mechanism. Using WGS to predict the antibiotic resistance has demonstrated sensitivity and specificity of 96% and 97% respectively, compared to the phenotypic detection assays (*Goldberg et al., 2015*).

## CONCLUSIONS

Complete characterization of the phenotypic and molecular mechanism of the panel strain will hold a great importance in pharmaceutical industry during the initial screening to evaluate the adequacy of antimicrobial drugs. The efficacy of the drug can be verified on pathogens displaying different resistant profiles, hence enabling their role before entering into clinical trials. These strains can also be used as standard reference strains and its

antimicrobial resistance profile can be used in laboratory settings. Additionally, this would improve our understanding of resistance phenotypes with its in-depth mechanism responsible for the same.

### Funding

This work was supported by the BioNano Health-Guard Research Center, funded by the Ministry of Science; ICT and Future Planning (MSIP) of Korea as a Global Frontier Project (Grant Number H-GUARD_2014M3A6B2060509) and the Ministry of Health & Welfare, Republic of Korea (grant number: HI14C1324). The funders had no role in study design, data collection and analysis, decision to publish, or preparation of the manuscript.

### Grant Disclosures

The following grant information was disclosed by the authors:
Ministry of Science; ICT and Future Planning (MSIP): H-GUARD_2014M3A6B2060509.
Ministry of Health & Welfare: HI14C1324.

### Competing Interests

Young-Lag Cho is an employee of LegoChemBioScience Inc., Daejeon, Korea.

### Author Contributions

- Roshan Dsouza conceived and designed the experiments, analyzed the data, wrote the paper, prepared figures and/or tables.
- Naina Adren Pinto performed the experiments, wrote the paper.
- InSik Hwang performed the experiments.
- YoungLag Cho contributed reagents/materials/analysis tools.
- Dongeun Yong conceived and designed the experiments, analyzed the data, wrote the paper, prepared figures and/or tables, reviewed drafts of the paper.
- Jongrak Choi analyzed the data, contributed reagents/materials/analysis tools.
- Kyungwon Lee conceived and designed the experiments, reviewed drafts of the paper.
- Yunsop Chong analyzed the data.

### DNA Deposition

The following information was supplied regarding the deposition of DNA sequences:
The draft genome sequences of isolates YMC2011/8/B10311, YMC2011/7/B774, YMC2013/7/B3993, YMC2011/7/B7207, YMC2011/11/B7578, YMC2010/8/B2027 and YMC2012/8/C631 have been deposited at DDBJ/ENA/GenBank under the accession LYPQ00000000, LYPS00000000, LDWV00000000, LYPU00000000, LYPT00000000, LYPV00000000 and LYPW00000000, respectively.

### Data Availability

The raw data has been supplied as a Supplementary File.

# Supplemental Information

Supplemental information for this article can be found online at http://dx.doi.org/10.7717/peerj.2896#supplemental-information.

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
