# Peer review of "Panel strain of *Klebsiella pneumoniae* for beta-lactam antibiotic evaluation: their phenotypic and genotypic characterization"

_PeerJ, doi:10.7717/peerj.2896_

## Round 0.1 · original submission · Major Revisions

As you can see, the report of the reviewer was quite complete and detailed. If you submit a revised version, you should explain how and to what extent you have taken each suggestion into consideration. Your revised version will most likely be resubmitted to this reviewer.

·

Basic reporting

The article fails on the area of resistance to antimicrobials

Experimental design

The experimental design of the article fails on genomics and phenotype characterization.

Validity of the findings

The article shows genomic sequencing, whose assembly needs to be better described.

Additional comments

Comments to authors
Manuscript entitled: “Panel strain of Klebsiella pneumoniae for beta-lactam antibiotic evaluation: Their phenotypic and genotypic characterization” by Roshan DSouza, Naina Pinto, InSik Hwang, YoungLag Cho, Dongeun Yong, Jongrak Choi, Kyungwon Lee, Yunsop Chong.

Reviewer's report:
In the manuscript (MS) entitled “Panel strain of Klebsiella pneumoniae for beta-lactam antibiotic evaluation: Their phenotypic and genotypic characterization” by Roshan DSouza et al., authors investigated the relationship between resistance phenotypes with its corresponding genotypic profile in several K. pneumonia (KP) isolates from 2009 to 2013 dispersed in only one hospital in Seoul, Korea.

I recommend Major revision, as follows:

Introduction
Line 54: In this case, I suggest change this sentence “production of high- level AmpC β-lactamase and penicillinase, acquisition of ESBL genes or carbapenemase genes, change in the membrane permeability or the efflux systems” for this one: “high-level production of AmpC β-lactamase and penicillinase, acquisition of genes encoding for Extended Spectrum Beta-Lactamase (ESBL) or carbapenemase, change in the membrane permeability or high-level expression of efflux pump systems”.

Lines 54 – 55: Before talking about the field of Massive parallel sequencing (MPS), the authors should include a paragraph related to the findings obtained from the Klebsiella pneumonia complete genomes. Currently there are several complete genomes of KP available in databases with the corresponding articles describing genes encoding for resistance factors, such as the authors are reporting in this manuscript. So, I recommend that the authors add such a paragraph, between the lines 54 and 55 of this version of MS.

Lines 61 to 63. Regarding this sentence: “This study was primarily designed to fulfill the need of pharmaceutical industry to evaluate the efficacy of novel antimicrobial compounds on K. pneumoniae expressing different resistance phenotypes”. In my opinion, the authors should place a narrower objective for this work, rather than a possible future breakdown from this study. This is also because of bacterial sample from this study is limited to a small geographical region. This work is most useful for hospital diagnosis to treatment with targeted antibiotics. The results of this work are still far to fail directly in the pharmaceutical industry, for which already other approaches are being applied, such as system biology including druggability score in combination with metabolic and regulatory network reconstructions.

Material and Methods

Line 68, 76 and others throughout the text. Be careful when you use the term “strain”, for example: “Bacterial strains”….. and…. “More than 2,000 K. pneumoniae strains”, be careful you must be treated these as isolates, rather strains.

Line 72. This sentence does not sound too fine: “These strains were screened in-silico for specific phenotypes”. Here, it is better just to say: “These strains were screened for specific phenotypes ….”.

Line 89. For this sentence: “minute at 6,000rpm and the supernatant was used as a genomic template”, change “genomic template” for “DNA template”.

Line 104 and others, change Iontorrent PGM → Ion Torrent PGM

Results

Firstly, in the results section are missing a table containing the assembly statistics data and the annotation features obtained from the analysis of genome sequencing with Ion Torrent discussed in the paper. This table can be included as an additional material and should be discussed in a sub-section of results.

Secondly, the authors must deposit the genomic sequences and CDS into a primary database, such as Genbank NCBI and add the corresponding access numbers in the manuscript.

Regarding Table 2. Authors must clarify which database were used to extract the information for sequence annotation of genes shown in this table. For example, what are the sequence differences between the SHV-1 gene, SHV-11, SHV-36, etc. A such database exist: http://www.lahey.org/Studies/, check polymorphisms of each gene and discuss showing a query sequence for the ones shown in Table 2 in a supplementary material.

Line 123. For the sentence: “Among total 3000 K. pneumoniae collection”. Better standardize the amount of the samples referred in Material and Methods and Results. In Material and methods, the authors are referring to ‘more than 2,000’.

Line 125. The authors should explain the term "wild type" in this context.

Line 126 and 133. standardize the term High-level or High level

Lines 137-139. Be careful with this discussion: Resistance to the piperacillin, ampicillin and ampicillin-sulbactam are due to hyperproduction of penicillinase TEM-1and SHV-11 beta- lactamase. The authors just have displayed the genomes sequences harboring several genes enconding for a putative phenotypes, they are not showing any gene expression nor enzyme production. So, the discussion throughout this section should be consistent and must be appropriate to the evidence shown from your data.

Line 139 – 140. For this sentence: “while OmpK36 gene expression was truncated or terminated due to the mutations present as observed in the WGS.” For this case, the authors must illustrate an alignment between two sequences: one without mutations in the sequence and the other one with mutations. This must be done for both situations, for mutations in the promoter region of the gene and also for mutations within the coding sequence. The figure can be included along with the supplementary material SDS-PAGE. The same must be perform for the result described in line 171: “However, the OmpK35 gene has been interrupted by IS1, thus providing resistance to cefoxitin”. But for this last sentence (OmpK35), the authors also should add a reference correlating this type of mutation with the resistance phenotype.

Lines 159-160. For the sentence: “This multidrug-resistant strain was unique because it contained multiple copies of ESBL gene.” Add the gene name in parenthesis.

Lines 162 to 164: In-depth analysis of the strain revealed 2 copies of bla SHV-12 and one copy of bla CTX-M-15 genes. In-addition, we also found one copy of bla OXA-9 and three copies of bla TEM-1 . How authors confirmed this gene copy number? in this case it is important that it be argued that this result is not an artefact's assembly. A such high copy number for these genes is expected? there are some other evidence of this in the literature?

Line 170 and others. Check Insilico → in silico.

Conclusion

Line 231 – 233. Again, the authors should be change the focus of this work, as I mentioned before for lines 61 to 63.

---

## Round 0.2 · accepted · Accept

The revision is satisfactory.

·

Basic reporting

"No Comments".

Experimental design

"No Comments".

Validity of the findings

"No Comments".

Additional comments

In this revised version the authors satisfactorily met each of the points that were asked. I believe the revised version is now consistent for publication.